# Strategies to Circumvent Host Innate Immune Response by Hepatitis C Virus

**DOI:** 10.3390/cells8030274

**Published:** 2019-03-22

**Authors:** Tapas Patra, Ratna B. Ray, Ranjit Ray

**Affiliations:** 1Departments of Internal Medicine, Saint Louis University, St. Louis, MO 63104, USA; tapas.patra@health.slu.edu; 2Departments of Pathology, Saint Louis University, St. Louis, MO 63104, USA; rayrb@slu.edu; 3Molecular Microbiology & Immunology, Saint Louis University, St. Louis, MO 63104, USA

**Keywords:** hepatitis C virus, innate immunity, interferon, cytokine, NK cell, complement

## Abstract

Innate immune responses generate interferons, proinflammatory cytokines, complement activation, and natural killer (NK) cell response. Ultimately, this leads to the induction of a robust virus-specific adaptive immunity. Although the host innate immune system senses and responds to eliminate virus infection, hepatitis C virus (HCV) evades immune attack and establishes persistent infection within the liver. Spontaneous clearance of HCV infection is associated with a prompt induction of innate immunity generated in an infected host. In this review, we have highlighted the current knowledge of our understanding of host–HCV interactions, especially for endogenous interferon production, proinflammatory response, NK cell response, and complement activation, which may impair the generation of a strong adaptive immune response for establishment of chronicity. The information may provide novel strategies in augmenting therapeutic intervention against HCV.

## 1. Introduction

HCV was identified in 1989 as one of the etiological agents of viral hepatitis [1]. HCV is a small enveloped RNA virus belonging to genus *Hepacivirus* and family *Flaviviridae*. The positive-sense single-stranded RNA genome of HCV is around 9.6 kb in size, flanked by UTRs on both ends. The single open reading frame present in this genome encodes for a nonfunctional polyprotein precursor which is cleaved by cellular and viral proteases into three structural and seven nonstructural proteins [2,3]. HCV is classified into seven genotypes and several subtypes expanding the diversity of the genomic sequence [4]. Approximately 177.5 million people in the world are infected by HCV, and annually, 1.3–3.7 million new cases of HCV infection are estimated [5,6]. HCV is spread parenterally, and its mode of transmission includes blood to blood, mother to baby, sexual, and organ transplantation. The clinical picture of HCV infection is often an acute stage, followed by chronic infection in about 80% of people initially infected [7,8]. The acute stage of infection may show mild flu-like symptoms, but is often asymptomatic. Chronic HCV infection often progresses to cirrhosis and hepatocellular carcinoma (HCC). Current estimates suggest approximately 5% of chronic HCV-infected patients may develop HCC [9]. HCC is the fifth most common cancer worldwide, and the third most common cause of cancer death [10] and liver transplantation in the United States [11].

An HCV-specific standard therapeutic regimen of pegylated interferon-α and ribavirin was approved by the Food and Drug Administration (FDA) in 1999. However, their use is limited to HCV patients with a single subset of genotype, prolonged treatment period, and severe side effects [12,13]. Recent introduction of direct acting antiviral (DAA) achieved sustained virological response (SVR) in a majority of HCV-infected patients [14]. However, transient virus clearance and rapid surge of viral resistance against these compounds may cause additional problems. There is a strong necessity for protective vaccine against all the HCV genotype variants, and it remains a challenge to control HCV infection globally [15].

The human immune system has developed two arms—innate and adaptive immunity—to act cooperatively, protecting against infection and limiting the damage caused by invading pathogens. Innate immunity acts immediately following infection, directing production of proinflammatory cytokines and orchestrating adaptive immunity. HCV has evolved mechanisms to evade host innate immune response for viral persistence. The persistence of HCV leads to chronic infection which ultimately advances liver disease [16]. It is important to fully interpret the immunopathogenesis of HCV infection and, eventually, exploit effective strategies to eliminate HCV. In this review, we discuss the evasion mechanisms of innate immune responses by HCV, which will deepen our understanding of the therapeutic intervention strategies used for immunity and liver disease prevention.

## 2. Endogenous Interferon Production

Interferon (IFN) pathways are tightly regulated by the host in a cell-intrinsic manner. Host cells are activated to produce type I IFN and proinflammatory cytokines from recognition of viral components, and upregulate a family of IFN-stimulated genes (ISGs) that exert inhibitory effects on viral replication [17]. HCV develops multiple strategies to escape or overcome the antiviral actions of IFN and make chronic infection in the host [18].

Host cells during acute virus infection respond through pathogen recognition receptors (PRRs) and recognize viral pathogen-associated molecular patterns (PAMPs). The retinoic acid inducible gene-1 (RIG-I)-like receptors (RLRs) are cytoplasmic RNA helicases that function as PRRs for the recognition of HCV RNA following infection. RIG-I has three major domains: the C-terminal repressor domain, central DEAD box helicase domain, and caspase activation and recruitment domain (CARD) at the N-terminal. The most important C-terminal domain of RIG-I selectively binds to the 5′-triphosphate, a distinguishing feature of non-self RNA. The 5′-triphosphate of the polyuridine core of the HCV RNA recognizes RIG-I and promotes conformational changes. This conformational change activates type I and type III IFN production by triggering innate antiviral immunity to HCV infection [19,20]. On the other hand, conformational changes in RIG-I cause interaction of mitochondrial-associated endoplasmic reticulum membrane (MAM) with mitochondrial antiviral signaling protein (MAVS). This interaction results in assembly of a signalosome complex that activates effector molecules, including the interferon regulatory transcription factor 3 (IRF3) and NFκB, to drive downstream innate immune signaling [21].

To understand how HCV antagonizes IFN signaling, the involvement of several viral proteins has been studied. HCV E2 and NS5A proteins interact with double-stranded (ds) RNA-activated protein kinase R (PKR) and disrupt PKR functions [22,23]. The HCV Core protein induces suppressor of cytokine signaling 3 (SOCS3) and SOCS1 expression, which blocks STAT1 function [24]. HCV Core and NS5A proteins suppress STAT1 phosphorylation in hepatocytes [25]. However, dephosphorylated STAT1, which accumulates in response to IFNs, maintains or increases the expression of a subset of ISGs independently of tyrosine-phosphorylated STAT1 [26]. We reported that HCV-infected hepatocytes display upregulation of total STAT1 without detectable phosphorylated STAT1, and modestly activate interferon-stimulated response element (ISRE) promoter [27]. In the early phase of HCV infection, the HCV NS3/4A protein cleaves MAVS and fails to transduce the RIG-I/MDA5 signal for IRF3-IFN-β activation [28,29]. Another study showed that interference of mitochondrial fission significantly increases ISRE activities, suggesting that HCV evolved strategies independent of NS3/4A in modulating innate immune responses [30]. 

HCV genotype 1b-infected Japanese patients revealed that sequence variation within HCV NS5A protein at the interferon sensitivity-determining region (ISDR) could predict IFN treatment outcome [31]. However, this observation remains debatable for lack of sufficient clinical observations. The capacity of IFN-α production in HCV-infected patients varies, since both high and low IFN-α levels have been reported [32]. IRF7, one of the key ISGs, plays a major role in IFN-α production. IRF7 undergoes phosphorylation when activated and translocates into the nucleus. IRF7 amplifies the type I interferon response by inducing expression of IFN-α, which also acts in both autocrine and paracrine manners through the IFN-α/β receptor. IRF-7 remains localized in the cytoplasm of HCV-infected hepatocytes [27]. An upregulation of IFN-α occurs at an early point of HCV infection in cell culture. The initial burst of IFN expression may be for uncoating the virus genome, RNA replication, and translocation of dephosphorylated STAT1 for ISREs activation. However, this activation is not sufficient to trigger high enough antiviral responses to clear HCV. Due to virus infection, IRF-7 fails to translocate into the nucleus and inhibits IFN-α synthesis. HCV infection also inhibits interferon-stimulated gene factor 3 (ISGF3) complex formations by targeting protein phosphatase 2A (PP2A) [33]. Genome-wide association studies (GWAS) suggested a strong relation of genetic variants near the interleukin 28B (IL-28B or IFNλ3) with pegylated IFN-α treatment-induced clearance of HCV [34]. Polymorphisms in the IL28B (*IFNL3*) and/or *IFNL4* gene also influence immune responses, the capacity to spontaneously eliminate HCV, and response to IFN therapy [35]. However, the current DAA regimen is interferon free, and does not influence genetic variation of the *IFNλ3* gene.

Toll-like receptors (TLRs) are germline-encoded molecules and are the key components of the innate immune system which recognize endogenous danger-associated molecular patterns (DAMPs) and exogenous PAMPs. Activation of TLRs may limit replication of infectious agents. The role of TLRs in chronic HCV infection has been reported [36]. In-vitro studies indicated that TLR2, -3, -4, -7, and -8 recognize HCV components as PAMP ligands. HCV Core and NS3 proteins trigger the TLR2 signaling pathway and activate inflammation [37]. On the other hand, both TLR3 and TLR7 play roles in sensing of HCV RNA. TLR3 is expressed in liver cells (hepatocytes and Kupffer cells) from HCV infection. TLR3 signals are transduced through the TLR domain containing adapter-inducing IFN-β (TRIF) leading to activation of the transcription factors IRF3 and NFκB for induction of innate immunity [38,39]. TLR4, a lipopolysaccharide receptor, plays a critical role in PAMPs and activation of innate and adaptive immune responses. HCV NS5A protein plays a potential role in resistance to IFN-α treatment by transactivating TLR4 promoter in vitro. TLR signaling is mediated by the adaptor protein myeloid differentiation factor 88 (MyD88), which triggers the activation of transcription factors important for proinflammatory cytokines. HCV NS5A also associates with the death domain of MyD88 and inhibits TLR7 signaling in mouse macrophages [40]. Further, HCV sensing by TLR7 occurs in both plasmacytoid dendritic cells (pDCs) and Kupffer cells, leading to production of IFN or activation of inflammasome (a multiprotein complex which plays a role in the innate immune response [41]. TLR7 and TLR8 share a high degree of structural similarity and variations in these genes, and impair immune responses during HCV infection [42,43]. Both activation and suppression of TLRs may be necessary to strengthen the anti-HCV immune response for limiting virus replication. Thus, the status of TLR signaling defines the type and strength of the anti-HCV immune response, and the outcome of infection. HCV interferes with the IFN signaling pathway at many different levels for establishment of persistent infection, and targeting these signaling molecules may provide additional therapeutic modalities. 

## 3. Induction of Proinflammatory Responses

Inflammation ensures the repair of damaged tissue and removal of detrimental stimuli by host cells [44]. Immune cells, such as macrophages and dendritic cells (DCs), are not the only ones playing an important role; the nonprofessional cells also contribute to inflammation induced by microbial infection [17]. Interleukin-1β (IL-1β) and IL-18 have important roles in combating the invading pathogen as part of the innate immune response. IL-1β-activating platforms, known as inflammasomes, assemble in response to pathogen-associated danger molecules. The inflammasome comprises a family of cytoplasmic membrane-bound PRRs collectively known as NOD-like receptors (NLRs) to sense viral nucleic acid and/or viral proteins. Once activated, NLRs form a multiprotein complex with apoptosis-associated speck-like protein containing a carboxy-terminal CARD (ASC) and caspase-1 for inflammasome assembly, which in turn activates caspase-1 [45]. The production of IL-1β and IL-18 is a tightly regulated process which requires two distinct signals for activation and release [46]. The first signal leads to NFκB activation and synthesis of pro-IL-1β and pro-IL-18 mRNA in a TLR signal-dependent manner. The second signal involves activation of caspase-1, which cleaves pro-IL-1β and pro-IL-18 into biologically active forms. High plasma IL-18 level was seen in the acute phase of HCV infection [47]. The status of IL-1β/IL-18 in HCV-infected hepatocytes and their induction through cross-talk with macrophages were studied meticulously [48,49,50].

We and others have reported that HCV induces secretion of IL-1β/IL-18 in the THP-1 cell line (a macrophage cell-culture model), human PBMC-derived macrophages, and primary human Kupffer cells (liver-resident macrophages). We have shown that the induction of these proinflammatory cytokines occurs via the NFκB signaling pathway, suggesting that HCV initiates inflammasome signal 1 pathway in macrophages. Subsequent studies demonstrated that HCV p7 RNA is sufficient to induce IL-1β secretion from macrophages and is inhibited by KCl or ion-channel blocker amantadine [48]. Pretreatment with a potassium-channel inhibitor in HCV-incubated macrophages reduces IL-1β maturation [49]. HCV poly(U/UC) RNA transfected into macrophages also triggers IL-1β mRNA expression and secretion, suggesting that HCV employs multiple strategies for triggering IL-1β secretion. Induction of IL-1β/IL-18 may have positive or negative influence on hepatic inflammation and disease outcome. Interestingly, our recent study demonstrated that HCV induced IL-1β from macrophages does not induce inflammation or activation in human hepatic stellate cells [51]. HCV does establish chronic infection, suggesting that IL-1β/IL-18 may not have an impact on HCV replication and virus clearance, although we cannot rule out their role in spontaneously cured infection.

During the acute-phase inflammatory response, TNF-α, along with other cytokines, is produced to activate endothelial cells and leukocytes. They influence the function of other cells involved in adaptive immune responses. In the liver, macrophages and Kupffer cells are the main sources of TNF-α [52]. The liver comprises mostly hepatocytes (~60–80%) which can produce TNF-α during chronic HCV infection [53,54]. The TNF-α signal is transduced by activation of transcription factor NF-kB that results in activation of various genes involved in cell proliferation and death, inflammation, and cancer [55]. Studies have shown that TNF-α promotes hepatocyte proliferation rather than their death when it is administrated to animals or incubated with hepatocytes in vitro [56]. The HCV NS3 protein inhibits TNF-α-induced NFκB activation via binding to linear ubiquitinin chain assembly complex (LUBAC) [57]. On the other hand, IL-6 plays a central role as a distress cytokine during inflammation in the body. During HCV infection, the production of IL-6 and IFN-β from B cells increases. These events follow a strong activation of TLRs, especially TLR4 which activates NFκB, and triggers IL-6 production by binding the promotor region of the IL-6 gene [58]. IL-6 can help self-production through binding to a heterodimeric IL-6R/gp130 complex. This heterodimer triggers an activation process and the JAK-STAT3 pathway retains the cell cycle progression [59]. It has been reported that HCV induces inflammation through a significant increase in serum IL-6 levels in chronically infected patients [60]. In addition, HCV infection may actively promote the development of hepatic steatosis via the paracrine effect of secreted IL-8 [61]. These proinflammatory cytokines lead to the development of chronic inflammation for HCV persistence in infected cells. HCV-induced inflammation through IL-6 activates JAK-STAT3 to retain cell cycle progression. 

Cells of the innate immune system, such as the dendritic cell (DC), may be poor stimulators of T cells. On the other hand, E2-treated macrophages significantly decrease expression of M1 phenotype markers (IFN-γ, IL-6, and TNF-α) as compared with untreated macrophages. In contrast, increased expression of M2 markers (MRC1 or CD206, IL-10, TGM2) is observed in HCV E2-treated macrophages [62]. HCV antigens may regulate inflammatory regulators differently based on the cell types and may be a mechanism by which HCV target cells impair development of a strong adaptive immunity.

## 4. Modulation of Natural Killer Cell Response

Natural killer (NK) cells are a large proportion of the granular lymphocyte population in the human liver, remain in a functionally hyporesponsive state, but rapidly induce an innate immune response to viral infection [63,64,65]. NK cells either directly target infected hepatocytes, or act indirectly by influencing other immune cells such as DCs or T cells for virus clearance. The interaction between NK cells and HCV-infected hepatocytes may result in the regulation of NK cell activity. There is an activation of NK cells in the acute phase of HCV infection indicating a role in the innate immune response. Genetic studies reveal that interaction between human leukocyte antigen HLA-C and specific killer immunoglobulin-like receptor KIR2DL3 results in spontaneous cytotoxicity of NK cells in HCV infection [66,67]. Indeed, myeloid dendritic cells (mDCs) produce IL-12 in response to HCV-mediated TLR-3 signaling and induce IFN-γ secretion by NK cells [68,69]. pDCs sense HCV RNA in exosomes generated from the infected hepatoma cells and secrete IFN-α which activates NK cells [41]. On the other hand, chronic HCV-infected patients have shown perturbations in NK cell frequency and function [70]. NK cell frequencies in peripheral blood are reduced in chronic HCV infection when compared to healthy individuals. IFN-γ is the major cytokine that NK cells secrete and is a critical factor for inhibition of viral replication. Concurrent engagement of activating receptors and cytokine receptors on NK cells induces IFN-γ secretion. Therefore, a decrease in activating receptor expression would likely be correlated with IFN-γ inhibition by CD56^dim^ NK cells [63]. A polarized NK cell phenotype is induced by chronic exposure to HCV-induced TFN-α. This phenotype may contribute to liver injury through TRAIL expression and cytotoxicity, whereas the lack of increase in IFN-γ production may facilitate the inability to clear HCV [71,72].

HCV E2 protein binds to the NK CD81 receptor, decreasing the release of IFN-γ and cytotoxic granules [73,74]. NKG2D from NK cells interacts with hepatocyte major histocompatibility complex class I-related chains A and B (MICA/B) as one of the ligands [75]. HCV affects NK cell activity through direct cell-to-cell interaction via CD81 or NK cell receptors or in an indirect manner via cytokine or TRAIL release [73,76,77,78]. NK cells exposed to cell culture-grown HCV-infected hepatocytes become unable to increase complement synthesis due to inhibition of MICA/B protein expression [79]. Further, NKG2D expression is lowered in circulating NK cells from patients with chronic hepatitis C [80]. A remarkable increase of hepatic NK cells for expansion of resident liver NK cells and/or recruitment of NK cells from the blood occurs during infection [81]. HCV NS5A protein stimulates monocytes through TLR-4 and induces secretion of IL-10, which subsequently stimulates the secretion of transforming growth factor (TGF)-β and downregulates NKG2D on the NK cell surface, resulting in functional impairment of NK cells [82]. HCV NS2 and NS5B proteins are also responsible for HCV-associated decrease in MICA/B, resulting in a loss of the C3/C4 complement components. This inactivation of the complement system leads to impairment of NK cell activation and attenuates adaptive immune response [79]. Furthermore, the complement system indirectly promotes dendritic cell-mediated NK cell activation by inducing TGF-β1, and involves the inflammatory response [83]. Together, this information suggests that the functional impairment of NK cells is associated with the evolution of HCV chronicity. 

## 5. Activation of Complement System

The complement system is a series of plasma proteins which work with the innate immune system for targeting and eliminating the invading pathogens. The complement plays a prominent role in the linkage of innate and adaptive immunity. The liver is the main source of complement and hepatocytes are the primary sites for synthesis of complement components in vivo [84]. Liver damage may diminish capacity of complement synthesis in patients. HCV successfully escapes the complement response for persistent infection by regulating complement components. C1q is the initial component of the classical complement system and plays a protecting role in viral infection. The binding of C1q to the C1q receptor, gC1qR, plays a role in resolution of infection [85]. HCV Core protein interacts with gC1qR which can cause pathogenic effects in vivo. The receptor circulates as a complex with HCV Core protein and sequesters the C4d cleavage to prevent complement-mediated lysis [86]. The gC1qR expression on CD4^+^T cells influences the outcome of HCV infection [87]. HCV increases the frequency of gC1qR^+^CD4^+^T cells, and viral persistence maintains this frequency at the late phase of infection, while individuals resolving HCV infection do not. HCV has evolved mechanisms to evade immune activation, including complement response. Serum C3 level is depleted in HCV-infected cirrhotic patients [88]. Dumestre-Perard et al. [89] reported that C4 activity is significantly lower and the level of different complement components is depleted in patients with chronic HCV infection. Our study revealed that HCV E2 envelope glycoprotein suppresses C3 expression and impairs macrophage and DC maturation for antigen presentation and normal CD4^+^ T cell stimulation [90]. HCV suppresses C3, C4 and C9 complement component synthesis and impairs membrane attack complex (MAC) [91] for attenuation of MAC-mediated microbicidal activity (Figure 1). HCV induces CD55/DAF as a negative regulator of complement activation. CD55 inhibits the formation and dissociation of C3/C5 convertases. C3 complement component is an important mediator of the humoral and T-cell immune responses [92,93,94,95]. C3 facilitates antigen uptake/presentation and immune cell priming. Receptors interacting with various activation fragments of C3 are expressed on a wide variety of cell types associated with immune function. C3 activity is required for optimal expansion of CD8^+^T cells during a systemic viral infection [96]. HCV Core protein upregulates CD55 expression on the cell surface and inhibits CDC [97]. Some tumors do not solely express a single variant of CD55, but also express different isoforms of the protein [98]. Thus, the induction of cell-associated and secretory CD55 expression by HCV infection in immortalized human hepatocytes may limit complement-mediated damage of infected cells and the cell-associated microenvironment. Antibodies against cancer cell surface proteins enhance ADCC and CDC [97,99]. Our observations suggest that tumorigenic immortalized human hepatocytes induced by HCV in culture become susceptible to complement- and antibody- dependent killing in the presence of CD55 blocking antibody [97]. These studies highlight the cooperative approach of HCV proteins in controlling of host NK cell and complement components or complement-associated CD55 protein functions to perpetuate virus fitness. Together, the information suggests a multifaceted approach undertaken by HCV to impair complement functions.

## 6. Innate Immunity Helping Adaptive Immune Response

NK cells constitute a bridge between innate and adaptive immune responses. NK cell-mediated DC activation interplays in priming the adaptive immune response. During acute HCV infection, DCs of the host should interact with the viral proteins to contribute the CD4^+^T and CD8^+^T cell responses for clearance or persistence of infection. DCs expressing HCV Core or NS3 protein show an impaired antigen presentation and maturation, which renders DCs unable to trigger T-cell activation [100]. mDCs generated from HCV-infected patients display a reduced CD86 and/or IL-12 expression, and impaired stimulatory potential against allogenic CD4^+^T cells [101]. In addition, mDCs from chronic hepatitis C patients exhibit a high cytotoxic activity and can kill T cells through the TRAIL pathway [102]. Additionally, an increased plasma level of HCV Core protein in chronic HCV infection causes less IFN-α production due to a reduced frequency of circulating pDC [32]. Mucosal-associated invariant T (MAIT) cells represent a conserved subset of T cells and participate in the innate immune response for protection against infection [103]. Recent studies have shown that in chronic HCV infection, the number of MAIT cells significantly decreases, and residual MAIT cells seem to suffer from immune exhaustion and senescence, which would contribute to the diminished innate defense and facilitate HCV persistence for liver disease progression [104].

The CD8^+^T cell is not very effective in chronic hepatitis C patients. Impairment of CD8^+^T cell response was noticed in transgenic mice where the expression of HCV Core was directed to the liver, and followed through the suppression of IFN-γ, TNF-α, and granzyme B production [105]. Liver-infiltrating lymphocytes obtained from chronic HCV-infected patients suggest high levels of PD-1 and a low level of CD127 expression, resulting in a suppressed function of HCV-specific CD8^+^T cells [106,107]. These results suggest that regulation of the PD-1 pathway is essential for impairment of HCV Core-mediated T-cell responses. Robust and broad HCV-specific CD8^+^T-cell responses are critical for spontaneous viral resolution in acute infection, and help in maintaining by CD4^+^T cells. The HCV E2 protein and a short RNA fragment encoded by E2 have been found to hinder PD-1, cytotoxic T-lymphocyte antigen 4 (CTL-4) and Tim-3 expression, respectively [108,109]. Inhibition of PD-1, Tim-3, and CTL-4 pathways can cause imbalance between Th17 and Treg cells, which might lead to the failure of the HCV-specific CD4^+^T-cell response [110]. Several studies have reported that HCV can induce myeloid-derived suppressor cells, cytokines like IL-10 and TGF-β, resulting in a promotion of Treg development and suppression of CD4^+^T-cell function [109,111]. Interestingly, the HCV Core protein activates CD69 expression to promote B-cell proliferation and increases IgG and IgM production with diminished SOCS-1 signaling, followed by downregulation of MHC class II expression. In this way, HCV affects the antigen-presenting function of B cells, but not Ig production [112,113], and dysfunction of adaptive immune response allows establishment of the persistent HCV infection in a host. The modulation of immune functions by HCV is shown in a simplified cartoon (Figure 2).

## 7. Conclusions

HCV makes successful strategies to antagonize the host immune responses and often persists as chronic infection leading to life-threatening end-stage liver disease. In this review, we have discussed how the HCV antigen is sensed to the host by the innate immune system and would inhibit or prime the adaptive immunity. HCV modulates inflammatory responses in different cell populations and microenvironments within the host. DCs or macrophages are critical for antigen presentation, and regulation of these cells may impair adaptive immune response. Elucidating the mechanisms by which HCV fails to activate DC or modulate macrophage function and impair generation of a strong adaptive immunity will identify strategies for prevention of viral persistence. The foremost accomplishment of HCV research is the discovery of interferon-free DAAs with a sustained virological response. This advanced antiviral therapy is well tolerated and successfully works on pan-genotypes in eliminating HCV infection. However, recent reports suggest emergence of viral resistance against these therapeutic compounds [114]. This may allow continued HCV transmission in high-risk groups and resource-constrained settings due to limited surveillance. In addition, HCV infection often causes a silent disease and late diagnosis may lead to progression of advanced liver disease. Further, detection of HCV carriers, lack of immunity against reinfection, insufficient access to DAA therapy, uncertainty about the magnitude of viral resistance development, and continued risk for severe liver damage are the major hurdles to overcome. We also need to identify steps for augmenting immune responses, and developing a protective vaccine for HCV is an unmet medical necessity. Global eradication of HCV will not likely be possible without a robust vaccine.

## Figures and Tables

**Figure 1 cells-08-00274-f001:**
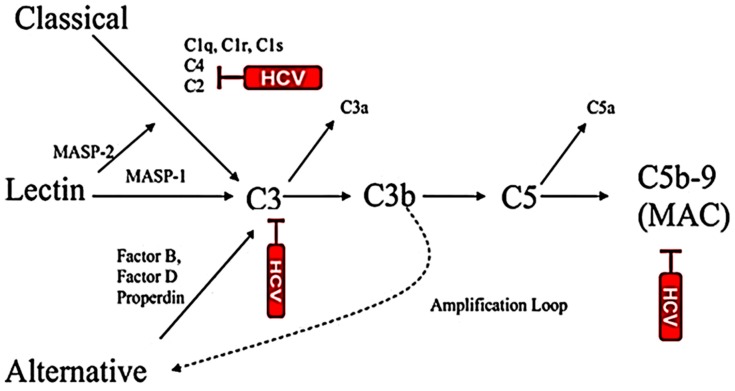
Inhibition of multiple complement components by HCV. Reduced complement function may exhibit attenuated MAC-mediated antimicrobial effect.

**Figure 2 cells-08-00274-f002:**
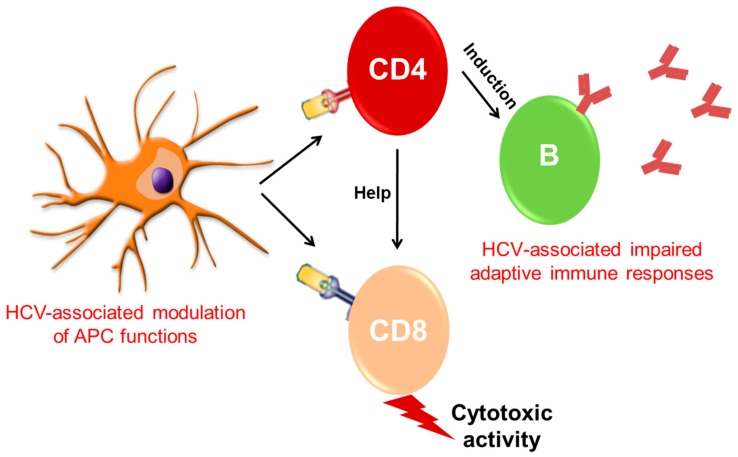
Modulation of APC and CD4^+^T/CD8^+^T-cell functions by HCV and impairment of adaptive immune responses.

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
