# Peer review of "Strategies to Circumvent Host Innate Immune Response by Hepatitis C Virus"

_cells, 2019, doi:10.3390/cells8030274_

Round 1
Reviewer 1 Report
1. In line 26, What is “UTRs”?
2. In line 42, delete “a single subset of genotype“.
3. In lines 45-46, please delete”However, the current…..especially for underdeveloped countries”.
4. Authors use “IRF3”, and so please make corrections from “IRF-7” to “IRF7”.
5. In abstract section, authors should mention about adaptive immunity.
Author Response
We have revised the manuscript based on all the 5 comments from the reviewer.
Reviewer 2 Report
Review of manuscript by T Parta, RB Ray and R Ray ”Strategies to circumvent host innate response by hepatitis C virus”
The review is a comprehensive overview of large bulk of information on the subject, with earlier as well as the latest studies on the subject. Results obtained by the authors are incorporated giving additional strength to the points and to overall conclusions.
The review would benefit from a number of modifications.
Scientific points to meet
Section 2 needs revision in TLR part
Lines 114-122 discuss HCV involvement and interference with TLRs and TLR signaling/TLR triggering of production of type IFNs. Currently, there is a statement that TLR3 and TLR7 sense HCV RNA (line 116, and again 119-121 for TLR7). Then where TLR3 are expressed, and how TLR3 signals are transduced. Respective information on TLR7 is very abrupt. No mentioning of other TLRs that are widely known to be involved and modulated in HCV infection (TLR4, TLR 8). After that, on line 121-122 follows statement “Thus, HCV interferes with IFN pathway in many different levels…” although how it interferes with respect to TRL signaling was not even mentioned.
The TLR subsection needs to be more structured.
Firstly, describe “switching on” of TLR signaling in response to HCV infection, which TLRs are involved.
Secondly, list genetic determinants of innate response efficiency, how differences TLR expression and signaling favor or disfavor HCV infection to confirm the role of TLRs in the infection course and outcome – like the studies done on TLR7, or TLR8 (TLR7 Wang CH et al, PLoS One 2011; https://www.ncbi.nlm.nih.gov/pmc/articles/PMC3192790/; TLR8 Fernández-Rodríguez A et al, 2015 https://www.ncbi.nlm.nih.gov/pubmed/26455634 ) through modulation of IFN-a production, and inflammation.
Thirdly, tackle regulatory aspects, how HCV infection modulates TLR signaling, and in this way, interferes in IFN pathway. For example, HCV antigens (as NS5a) increase expression of TLR4, hence of IFNb and IL6 inducing inflammatory response favoring chronic infection (Mchida K et al, 2006; https://www.ncbi.nlm.nih.gov/pmc/articles/PMC1346849/ ) (mentioned now on lines 166-167, but belongs here). At the same time, NS5A suppresses activation of IRAK-1 through its interaction with MyD88. This leads to the shutting off of TLR7 and TLR8 signaling, and to the suppression of maturation and differentiation of pDCs (Abe T et al, 2007 https://www.ncbi.nlm.nih.gov/pubmed/17567694 ).
A statement from lines 246-248 (HCV NS2 and NS5B proteins are also responsible for HCV-associated decrease in MICA/B, resulting in a loss of the C3/C4 components” would fit here very well.
After this, one can state that status of TLR signaling defines the type and strength of anti-HCV immune response, and outcome of the infection.
2. Section 3 needs complementation.
Line 156 starts with “During acute phase, inflammatory response, TNF-a along with other cytokines is produced to activate endothelial cells and leukocytes”. An explanation of how inflammatory response (in the acute stage) influences formation of the adaptive immune response – in norm – is needed. Then it would be easier to comprehend why switching off of TNF-a and switching on/increasing of IL-6 and IFN-b retains inflammation/favors chronic inflammation and impairs adaptive immune response.
Very important statement by the authors that HCV induced inflammation (through IL-6) activates JAK-STAT3 to retain cell cycle progression deserves to be in conclusions part for this section. It stresses not so much the role of chronic inflammation in suppression of specific immune response during disease chronization (as stated in currently in conclusive lines 173-174), but rather the need of chronic inflammation of HCV persistence in the infected cells. Ie innate responses are turned against the host and sustain chronic viral infection.
3. Section 4 on the complement needs extension and complementation
It would benefit a lot from coverage of complement-related issues in full, of HCV impairment of complement synthesis, activation and functions.
Components of complement are mixed and follow one after another without logics. Much attention is devoted to C3 (lines 186-191, 193-195). C4 comes in-between description of C3 involvement (lines 192-193). Lines 199-201 refer to several complement components – C3. C4, C5. Would be logical to start with papers describing interference of HCV with expression and activation of several C components, and then continue with single components.
Reference to M1 phenotype and phenotype markers inside complement section seems not motivated (lines 195-198).
4. Section 5 on modulation of NK cell response – would be more logical after section 3 in connection to the induction and maintenance of pro-inflammatory responses, before complement section.
5. Section 6 on innate immunity helping adaptive immune response
Subsection on complement in section 6 starting with “A prominent role of complement plays in the linkage of innate and adaptive immunity” (line 271) again refers to the synthesis of complement components, results of liver damage, and rol of HCV in regulation of synthesis of C3, C4 and C9. This part should be in the complement section, leaving here only the translation from complement system into the adaptive immune response.
In a way, this is a problem of these section that connects all arms of innate immune response, endogenous IFN production, proinflammatory cytokines, NK cells, complement to the adaptive immune response. It has to repeat statements about each of the components. May be more logical to use the information in this section, on how this element of innate immune immunity (or to rather disruption of the element) translates into deficiency in the adaptive immune responses, to complete each of the preceding section (2-5)?
Technical issues
A list of abbreviations would be highly appreciated.
Statements “Some tumors do not solely express a single variant of CD55, but also express different forms of the protein” (lines 202-203) needs reference. Same about the role of CDC and ADCC (lines 206-207).
Same is valid to authors own observations “that tumorigenic immortalized human hepatocytes induced by HCV in culture” (lines 207-209). Needs reference to authors own paper in “Virology” in 2000 showing immortalization by expression of HCV core https://ac.els-cdn.com/S0042682200902952/1-s2.0-S0042682200902952-main.pdf?_tid=8562e969-4a39-4aff-aeb9-72bb3be28a31&acdnat=1549295711_30111f68f6d0b35294179b9c675e219d and eventually to papers showing similar effect of HCV infection.
MAIT cells are mentioned – lines 265-266, description of their nature and role needs a reference.
Lines 292-293 – “HCV can induce MDSC…” – MDSC were not mentioned before, abbreviation needs to be deciphered.
Language corrections needed
Line 47-48 –“there is a paucity… and remains a challenge” – smth is missing (absence of vaccine or vaccine itself remains a challenge?)
Line 82 - “disrupt to catalyze” should be changed, could be “disrupt PKR functions (PKR-mediated catalysis)”
Line 105 – “virus infections fails to translocate IRF-7 into nucleus” – infection is not translocating. “Due to viral infection, IRF-7 fails to translocate into the nucleus”
Line 125 – “Immune cells ….not only play important roles; the nonprofessional cells also contribute..” – should be rewritten. “Immune cells are not the only ones playing an important role”
Line 145-146 – “hCV p/ RNA is sufficient to induce …. And inhibited by KCl…” part of the sentence is missing.
Line 158-159 – “Hepatocytes --- they only produce during chronic HCV infection” – they only produce what?
Line 176 – “The complement system is a series of plasma proteins work with the innate immune system” – smth is missing. Possibly, should be “The complement system is a series of plasma proteins which work with the innate immune system”.
Line 178 – “C1q is the initial components” – should be “C1q is the initial component”
Line 222 – “KIR2DL3 results spontaneous cytotoxicity” – should be “KIR2DL3 results in spontaneous cytotoxicity”
Lines 224-225 – “pDCs sense HCV RNA… and secretes IFN-a..”, should be “pDCs sense HCV RNA… and secrete IFN-a..”
Lines 243-244 “A remarkable increase of hepatic NK cells.. occur during infection” – should be “A remarkable increase of hepatic NK cells.. occurs during infection”
Lines 254-255 – “NK mediated DC activation engage in a key interplay” – should be “NK mediated DC activation engages in a key interplay”
Lines 260-261 – “… decreased secretion of IL12p70 followed by high level of IL-10, and ultimately causes anti-HCV T-cell responses” - what causes responses? mDCs – the “Cause response”. If it is decreases secrtion of IL12p70 and high level of IL-10, they CANNOT cause (not causes) anti-HCV response, rather on contrary. Needs clarification.
Phrase “A prominent role of compment plays in the linkage of innate and adaptive immunity” (line 271) is most grammatically wrong.
Phrase “Effectiveness of CD8+ T cell is defective” is wrong. Effectiveness cannot be defective. Process is either effective or not. Functions could be defective.
Some text is missing between lines 283 and 284.
Line 290-291 – “Obstruction of PD-1, CTL-4 and Tim-3 pathways imbalance between Th17 and Treg cells…” – something is missing “Obstruction of PD-1, CTL-4 and Tim-3 pathways, and imbalance between Th17 and Treg cells…”?
Author Response
Section 2- We have revised based on all the comments, excepting Lines 166-167 and 246-248. We did not feel comfortable to move those lines as they may not fit well in the other unrelated topics.
Section 3-We have revised the manuscript as the reviewer suggested.
Section 4 – We have revised the manuscript based on all the comments.
Section 6 – We revised the manuscript following reviewer’s suggestions.
Technical issues – All issues are addressed in the revised manuscript.
All language corrections are made as suggested by the reviewer.
Reviewer 3 Report
Manuscript ID: cells-430545; Patra et al.
This is a review on innate immunity in HCXV infection. It was written by authors with a good track record in the field in terms of scholarly publications. On the whole, the review is up to date but appears somewhat bloated (densely written) and disorganized. It would be markedly improved by including 1-2 figures. There are multiple issues with language. Please consider the following comments:
1. Page 1, line 10. « Host has evolved to induce innate immune responses… ». It’s not the host that induces the responses, it’s the pathogen.
2. Page 1, line 16. This sentence makes no sense. Please reformulate.
3. Page 1, line 20. « …targeting HCV associated tumor antigens… ». This topic is not addressed in this review to any significant extent.
4. Page 1, section 1, paragraph 1. This background section is largely generic. It will be useless to anyone with any knowledge of hepatitis C.
5. Page 2, line 44. « …achieve cure to a sustained virological response… ». Language issue.
6. Page 2, lines 47-48. « There is a paucity of protective vaccines… ». In fact, there isn’t any.
7. Page 2, line 74. Please define « IMFTs ».
8. Page 2, line 79. Reference no. 21 doesn’t seem to support this statement.
9. Page 2, 82. « …disrupt to catalyze… »???
10. Page 2, line 86. Please use the acronym « ISGs » consistently throughout the text. Also see Page 3, line 97 and 106.
11. Page 2, lines 89-90. « …cleaves MAVS and unable to transduce… ». Language issue.
12. Page 3, line 95. Why is it debatable?
13. Page 3, lines 100-101. Why is this interesting?
14. Page 3, lines 112-113. Why would a drug regimen influence genetic variation of the IFN lambda 3 gene?
15. Page 3, line 121. You have not yet told the reader what the inflammasome is.
16. Page 3, line 130. PRRs. Please use acronyms consistently.
17. Page 3, line 136. TLR. Please use acronyms consistently.
18. Page 3, lines 138-140. Reformulate this sentence which is hard to comprehend in its present form.
19. Page 4, line 150. Please delete this useless sentence.
20. Page 4, line 153-155. But there is the case of spontaneous HCV clearance, which occurs 25-30% of the time, no? Could IL-1beta/IL-18 have an impact on that?
21. Page 4, lines 158-159. They only produce… what?
22. Page 4, line 165. « …during inflammation in body… », « During hepatitis C infection… ». Language issues.
23. Page 4, line 180. « …may have an extremely important role in resolution of infection ». This would seem to be an overstatement based on the current state of knowledge in the field.
24. Page 5, line 202. Use the « CDC » acronym consistently and define it upon first use (also see line 206).
25. Page 6, line 259. The « mDCs » acronym was already used for myeloid dendritic cells (page 5, line 223).
26. Page 6, lines 270-280. This should go in the section on complement (section 4), shouldn’t it?
27. Page 6, line 284. Some words appear to be missing before « obtained ».
28. Page 7, section 7. Disappointingly, instead of bringing together the main issues that were raised/addressed in the review, the conclusion is merely a combination of generic statements on HCV and hepatitis C.
Author Response
We have extensively reorganized the manuscript and included two figures.
Comments 1-3: The manuscript is revised addressing these comments.
Comment 4: We agree with the generic nature of the background section. However, we felt that the readers of diverse interest may need this generic background.
Comments 5-28: Revisions are made based on all these comments from the reviewer.
Round 2
Reviewer 3 Report
The manuscript is improved. The authors have reworked the text and added two figures. Extensive editing of English language and style are required.
Author Response
Thanks for the comment. We have carefully edited the manuscript. Hope it reads much better.